# Physical and biological properties of a novel anti-adhesive punctate uneven gelatin film

Yuki Ozamoto[1,2☯], Tsunehito Horii[2,3☯]*, Shinichiro Morita[2☯], Hiroyuki Tsujimoto[2,3], Yasumitsu Oe[1], Hiroshi Minato[4], Joe Ueda[5], Hiroshi Ichikawa[2], Akihiro Kawauchi[3], Akeo Hagiwara[2,3], Susumu Kageyama[3], Masaji Tani[1]

1 Department of Surgery, Shiga University of Medical Science, Otsu, Shiga, Japan, 2 Division of Medical Life System, Department of Life and Medical Science, Doshisha University, Kyotanabe, Kyoto, Japan, 3 Department of Urology, Shiga University of Medical Science, Otsu, Shiga, Japan, 4 Department of Surgery, Yawata Central Hospital, Yawata, Kyoto, Japan, 5 Department of Gastroenterology, Ueda Clinic, Takanosu, Akita, Japan

☯ These authors contributed equally to this work.

* thorii@belle.shiga-med.ac.jp

**Data Availability Statement:** All relevant data are within the paper.

## Abstract

Since abdominal adhesion are quite problematic in abdominal and pelvic surgery, the conventional HA/CMC film are commonly used as an anti-adhesive material. However, such types are difficult to be rolled and delivered through the port of laparoscopic surgical devices due to adherence to the laparoscopic port or other parts of the films. To create an anti-adhesion film with more favorable handling properties and anti-adhesive effect, we developed a novel punctate uneven gelatin film (PU GF). In this study, we examined the physical strength, flexibilities and adhesiveness between film to tissues or film each other, compared to the conventional film and the flat gelatin film (Flat GF). In addition, we investigated the cell proliferation on each film and the anti-adhesive effect of the films and those reattachment possibility using a rat cecum abrasion model. The PU GF showed excellent tensile strength, ductility, and adherence to tissue compared to Flat GF and the conventional film. Moreover, the adherence of PU GF to the other film and to a silicon sheet were much lower than those of the Flat GF and conventional film. The proliferation of cells in PU GF and Flat GF were suppressed compared with control, though increased with time. The anti-adhesive scores of the PU GF after one time and re-attachment were significantly higher than that of non-covered control, although there was no significant difference between that of the conventional film and control. Our findings suggest that PU GF improve handling properties of laparoscopic surgery as it has excellent physical strength, ductility, and adherence to tissue, and low adherence to trocar. In addition, the punctate film may be more useful with the re-attachability without tearing and to retained sufficient anti-adhesion effect.

## Introduction

The frequency of post-operative adhesion is high in clinical settings, and it potentially causes bowel obstruction, female infertility, and chronic pain [1–9]. To prevent post-operative

**Funding:** The author(s) received no specific funding for this work.

**Competing interests:** None of authors have any conflicts of interest to declare.

adhesion, numerous anti-adhesive materials have been developed for clinical use. Hyaluronic acid and carboxymethyl cellulose film has been conventionally used (the conventional film) as an anti-adhesive material in clinical settings [10–15]. However, there are some unfavorable problems such as controversial clinical effectiveness to prevent adverse events of adhesion, and a potential risk of anastomosis leakage of intestine [16–20]. Also, in the recent study of Hyaluronic acid and carboxymethyl cellulose gel, the gel application reduces the number of organs involved in adhesions in an ischemic button model, but no overall reduction in adhesion formation was encountered [21].

Previously, we developed thermally cross-linked gelatin film (GF) as a novel anti-adhesive material, to solve these unfavorable problems [22–25]. Our previous study reported that the GF had better physical strength and ductility, significantly higher anti-adhesion effect with excellent peritoneal regeneration compared to the conventional film, without any cytotoxicity [24]. It can also be safe to use at an intestinal anastomosis without decreasing bursting pressure. Moreover, it can be reattached with high physical strength without tearing, has low stiffness that doesn't induce tissue injury, and a adequate anti-adhesive effect. In addition, GF does not act as a convenient scaffold for tumor cell growth and carries little risk of accelerating peritoneal dissemination when used as an anti-adhesion material in surgery for abdominal tumors [25].

On the other hand, such flat-types of films, including our GF are difficult to be rolled and delivered through the port of laparoscopic surgical devices due to adherence to the laparoscopic port or other parts of the films [26, 27]. Thus, any modification has been required in flat-types of anti-adhesion films. We developed a novel punctate uneven gelatin film (PU GF) as a unique surface form. Our PU GF was designed to decrease the contact area between the films themselves or between the film and the silicon sheet. In addition, the PU GF fits the tissue after moistening.

In this study, we examined the physical strength, flexibility, and adhesiveness to tissues or other films by conducting tensile, adhesive, and shear stress tests to compare PU GF with flat GF (Flat GF) and the conventional film. Furthermore, we investigated the cell proliferation on each film *in vitro* and the anti-adhesive effects of each film and their reattachment properties using a rat cecum abrasion model *in vivo*. Finally, the punctate gelatin film showed low adherence to other films and to silicon with high physical strength, ductility, adherence of tissue. In addition, the punctate film may be more useful with the re-attachability without tearing and to retain sufficient anti-adhesion effect.

## Materials and methods

### Materials

**Preparation of anti-adhesion films.** Flat GF and PU GF were supplied from Gunze Ltd. (Ayabe, Kyoto) and these methods of manufacturing are shown as follows. Gelatin solution of alkali-treatment was retrieved from porcine subcutaneous tissue (type I-collagen, Medigelatin®; Nippi Co. Ltd., Shizuoka, Japan) with 132000 Mw, gel strength at 257 g and an isoelectric point of 5. The gelatin solution is clinically safe, due to removing collagen telopeptides that induce bacterial endotoxins, major antigenicity, transmitting agents (prions) and contaminated viruses. The final concentration of gelatin solution was 4.5% with distilled water.

To make PU GF, the 10 ml of gelatin was poured on punctate plastic boards (Kanto Chemical Co., Tokyo, Japan) with mold release agent (10 cm x 10 cm) and dried on a draft for 48 hours. The PU GF was thermal cross-linking by vacuum dryer (AVO-250N, As One, Osaka, Japan) for 3.5 hours at 140˚C (86% ± 3% of the water content). The punctate structure was designed by pin-spot shape because of the industrial manufacturing method, as shown by the

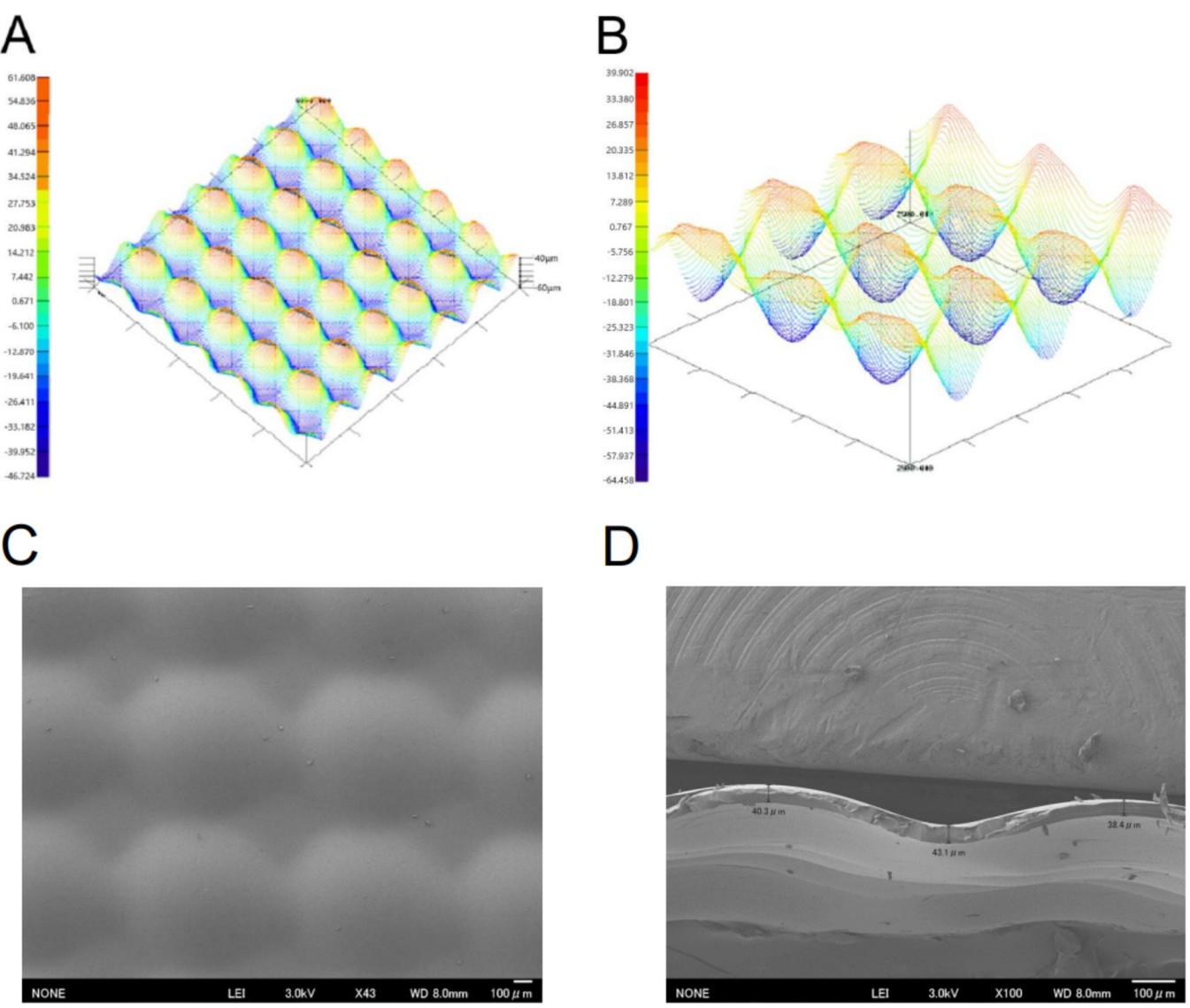

**Fig 1. The surface and cross-sectional structure of PU GF.** (a), (b) Surface roughness measurement (c) Surface SEM image (d) Cross-sectional SEM image.

patent of anti-adhesive material (WO2013/018864 A1). Surface roughness, SEM and cross-sectional SEM are measured in dry state (Fig 1). We confirmed that there was almost no change of the surface structure of the PU GF in wet condition just after moisture.

To make Flat GF, the gelatin was poured on flat plastic boards (Kanto Chemical Co., Tokyo, Japan) and dried on a draft for 48 hours. The Flat GF was thermal cross-linking by vacuum dryer (AVO-250N, As One, Osaka, Japan) (3.5 hours, 140°C, 86% ± 3% of the water content).

The conventional film (Seprafilm®; Genzyme Co., Cambridge, MA, USA) was used to compare with anti-adhesive materials. The film is a sterile, bioresorbable adhesive barrier with sodium hyaluronate and carboxy methyl-cellulose. Also, the polymers have chemical modification with an active agent 1-(3-dimethylaminopropyl)-3-ethylcarbodiimide hydrochloride in the information of product (http//www.genzyme.com).

All films have sterilization with ethylene oxide gas (SA-H160, elk Co., Osaka, Japan) (4 hours, 40°C, 0.43 g/L). Also, all films were saved in a dry condition in a desiccator.

**Animals.** Wistar/ST rats (Female, age of 7 weeks, weigh of 200 g) were purchased a week before the examination (Shimizu Animal Laboratory, Kyoto, Japan). All rats were raised in a specific-pathogen-free room with a 12-hour light-dark cycle, mean humidity of 50% and temperature around 23°C. Standard laboratory water and rodent chow were freely available. A beagle dog (Female, age of less than 2 years, weigh of 9.9 kg) was purchased (Shimizu Animal Laboratory, Kyoto, Japan). All animal housing, care and procedure of surgery abided with the guidelines of the Committee for Animal Research of Shiga University of Medical Science (ethical no. 2019-2-6), Doshisha University (approval ethical no. A15021), and Aya Co. (Shiga, Japan).

**Cells.** Rat fibroblasts were purchased from Lonza Ltd. (Basel, Switzerland). The cells were maintained in Dulbecco's modified Eagle's medium supplemented with 10% fetal bovine serum and kept in an incubator (37°C, 5% $CO_2$).

**Tensile test.** Each oblong piece (10×50 mm) of GF films was set to two folders of the apparatus of CPU gauge: MODEL-RX10, TESTSTAND: MODEL-1356R (Aikoh Engineering, Osaka, Japan) by gripping the films' ends (distance of 3 cm). Then, each film was pull into opposite directions automatically until the film was broken (5 mm/min speed). The calculations of the maximum stress (N/mm$^2$), fracture strain, and Young's modulus were conducted from the stress-strain ($\sigma_{nom}$ = f ($\varepsilon_{nom}$)) diagram by the following equations: (1) $\sigma_{nom} = F/A_0$ (2) $\varepsilon_{nom} = \delta/L_0$

where F was applied force; $A_0$, the first cross-section; $\delta$, the amount of change rate in gauge length; and $L_0$, the first gauge length. Six pieces of each film type were examined. Continuous variables are presented as mean ± standard deviation.

## Adhesive test

**Pull adhesive examination.** The examination was conducted by the testing machine of CPU gauge: MODEL-RX10, TESTSTAND: MODEL-1356R (Aikoh Engineering, Osaka, Japan). The upper stud was adhered to each film (10×10 mm) with double sided tape (Waki Co. Ltd., Osaka, Japan). Then, the lower stud was adhered to the abdominal wall from rats (20×20 mm) using a bonding agent (CEMEDINE®, CEMEDINE Co., Ltd., Tokyo, Japan). The two studs were attached and pressed together (12.5 KPa, 2 min). Then, each stud was pulled automatically in opposite directions at 5 mm/min speed. The test ended when each film of upper pull stud completely detached from the wall of abdomen. The maximum pull adhesive load was measured. The strength of pull adhesiveness was calculated in KPa. Six pieces of each film type were examined. Continuous variables are presented as mean ± standard deviation.

**Shear stress examination.** Large intestinal walls (length of 15 cm) were resected from three beagle dogs under intravenous anesthesia of pentobarbital (40 mg/kg of body weight) and inhalation anesthesia of sevoflurane (1–2%). The intestinal walls were set sideways on a horizontal-type force test apparatus of CPU gauge: MODEL-1016C, TESTSTAND: MODEL-2152VCE (Aikoh Engineering, Osaka, Japan). Each oblong piece (10×40 mm) of each film was grasped by one end of the film (distance of 1 cm) using the testing rig. The remaining 3 cm was then attached on the large intestinal wall for 2 min. The strength of film was measured parallel to the attached intestinal wall at 1 cm/sec speed until the film had sheared off completely and the maximum load was measured. Six pieces of each film were examined. Continuous variables are presented as mean ± standard deviation.

**Film-to-film attachment examination.** Two pieces of each film (10×50 mm) were attached to a metallic plate by double sided tape (Waki Co. Ltd., Osaka, Japan). Two metallic

plates with films were fixed by two folders of horizontal-type force test machine of CPU gauge: MODEL-1016C, TESTSTAND: MODEL-2152VCE (Aikoh Engineering, Osaka, Japan). A width of 10 mm from the end of each film was attached. The film was moistened using a spray with distilled water (5 cm apart from the film, five times pushing). The film tension was tested at 5 mm/sec speed, until the films separated completely. The maximum tensile load was measured, and six pieces of each film were measured. Continuous variables are presented as mean ± standard deviation.

**Film-to-silicon attachment examination.** A piece of each film (10×50 mm) was attached to a metallic plate by double sided tape (Waki Co. Ltd., Osaka, Japan). The metallic plate with film and a silicon plate were fixed by two folders of a horizontal-type force test machine of CPU gauge: MODEL-1016C, TESTSTAND: MODEL-2152VCE (Aikoh Engineering, Osaka, Japan). A width of 10 mm from the end of the film was attached to the silicon plate and the film tension was tested at 5 mm/sec speed until the films had separated completely. The maximum tensile load was measured, and six pieces of each film were examined. Continuous variables are presented as mean ± standard deviation.

**Cell proliferation test.** Testing films (15 mm in diameter) were set on the 24-well plate for culturing (CORNING®, NY, USA). The fibroblasts of rat were cultured, harvested and suspended in the DMEM (Wako Pure Chemical Industries, Ltd, Osaka, Japan). The $10^4$ cells in 750 μL of the cell suspension was poured to each well on only the plate (as control) or with each film. The viable cells in each well were counted using an ATPLite Kit® (PerkinElmer, MA, USA) at each 1, 3, 5, and 7 days after applying. Four wells per control or each film were measured at each day. The cell number was estimated with the ATP count from the preliminarily standard curve between the cell count and the ATP count. Continuous variables are presented as mean ± standard deviation.

## Anti-adhesive effect test

**Single-use anti-adhesive effect examination.** Thirty-two Wistar/ST rats were divided into four groups of eight rats each randomly: Flat GF, PU GF, the conventional film, and control group. Under intraperitoneal administration of pentobarbital (40 mg/kg of body weight) and inhalation anesthesia of sevoflurane (1–2%), the serosal side of the anterior wall of the cecum was abraded (15-mm-diameter) using a sanding tip for dental use (MINITOR Co., Tokyo, Japan) until a small blood drop was found. Another abraded region (15-mm-diameter), directly opposite the abraded cecum, was then made on the right lateral internal abdominal wall 2 cm apart from the middle incision on the abdominal wall. In the groups of films, the abraded cecum was covered with each oblong film (20×30 mm). No treatment was received in the group of control. Before closing the abdomen, both cecum and abdominal surfaces were sewn together with 6/0 Prolene® sutures (Ethicon Inc., Tokyo, Japan), inducing tight abdominal adhesion between both sites. Three weeks after the treatments, all experimental animals were sacrificed. The condition of the abrasion sites and abdominal cavity, including remaining material, was then checked by macroscopic observation. The numerical grade and score of adhesion were evaluated by extent and severity (Adhesive Scores in Table 1). The blinded assessment was performed by a researcher to the animal assignments (Fig 2). Continuous variables are presented as mean ± standard deviation.

**Anti-adhesion effect examination of re-attachment of film.** Thirty-two Wistar/ST rats were divided randomly into four groups of eight rats each: Flat GF, PU GF, the conventional film, and non-treated control group. Under intraperitoneal administration of pentobarbital (40 mg/kg of body weight) and inhalation anesthesia of sevoflurane (1–2%), the serosal aspect of the anterior wall of the cecum was then abraded (15-mm-diameter) using sanding tip for

**Table 1. The adhesive scores.**

| Category and Description | Score |
|---|---|
| (Extent) | |
| No Involvment | 0 |
| ≤25% of the site involvement | 1 |
| ≤50% of the site involvement | 2 |
| ≤75% of the site involvement | 3 |
| ≤100% of the site involvement | 4 |
| (Severity) | |
| No adhesion | 0 |
| Adhesions fall apart easily | 1 |
| Adhesions can be lysed with traction | 2 |
| Adhesions required <50% sharp dissection | 3 |
| Adhesions required <50% sharp dissection | 4 |

dental use (MINITOR Co., Tokyo, Japan) until a small blood drop was found. In the film groups, the film was attached to the serosal aspect of the anterior wall for 2 min. Then, the was detached from the wall and re-attached to the cecum abraded area. Another abraded region (15-mm-diameter), directly opposite the abraded cecum, was then made on the right lateral internal abdominal wall 2 cm apart from the middle incision on the wall of abdomen. No treatment was received in the group of control. Before closing the abdomen, both cecum and abdominal surfaces were sewn together with 6/0 Prolene® sutures (Ethicon Inc., Tokyo, Japan), inducing tight adhesion between both sites. Three weeks after the treatment, all experimental animals were sacrificed. The condition of the abrasion site and abdominal cavity, including remaining material, was then checked by macroscopic observation. The numerical grade and score of adhesion were evaluated by extent and severity (Adhesive Scores in Table 1). The blinded assessment was performed by a researcher to the animal assignments. Continuous variables are presented as mean ± standard deviation.

After evaluating the adhesion status, all animals were sacrificed by intraperitoneal administration of pentobarbital (100 mg/kg of body weight) and the segment with the abraded sites

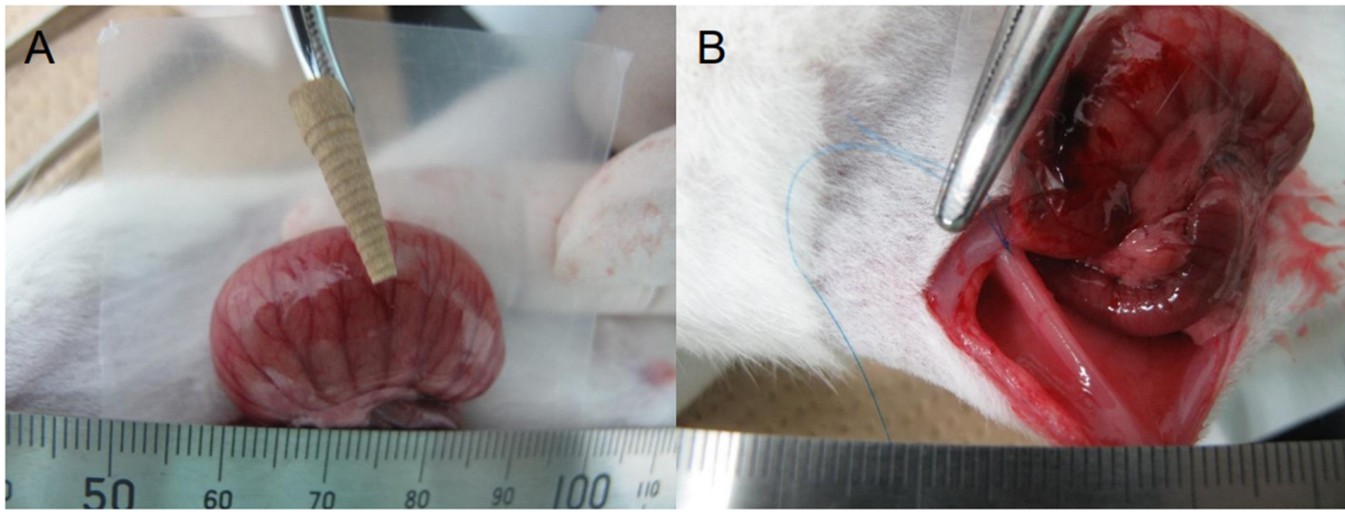

**Fig 2. Rat cecum abraded model in anti-adhesion tests.** (a) A surface of rat cecum and a sanding tip (b) Two abraded surfaces approximated with sutures.

was fixed with 10% buffered formalin and processed for embedding in paraffin. Transverse sections (3 mm) of the segments were stained with hematoxylin and eosin (H&E).

## Statistical analyses

Statistical comparisons of tensile and adhesive tests were made using analysis of variance test among all groups. Post hoc testing was performed by Scheffe test. Significant difference was shown in a $p$ value of $< 0.05$. Statistical comparisons of anti-adhesive effect tests were processed by Dunn's test following Kruskal Wallis test. Significant difference was shown in a $p$ value of $< 0.05$. Statistical comparisons of cell proliferation test were processed by Tukey's test as a post-hoc test after processing by one-way layout analysis of variance. Significant difference was shown in a $p$ value of $< 0.05$.

## Results

### Tensile test

The results of the tensile exam are shown in Fig 3. The maximum tensile stresses of the conventional film, Flat GF, and PU GF were $36.4 \pm 5.3$, $83.19 \pm 15.89$, and $73.83 \pm 13.35$ MPa, and the maximum fracture strains were $3.3 \pm 0.7$, $6.0 \pm 1.27$, and $7.27 \pm 2.5$%, respectively. The maximum tensile stress and fracture strains of Flat GF and PU GF were significantly higher than those of the conventional film ($p < 0.01$). Young's modulus of those films were $1.31 \pm 0.1$, $1.36 \pm 0.086$ and $1.16 \pm 0.14$ GPa, respectively. Young's modulus of PU GF was significantly

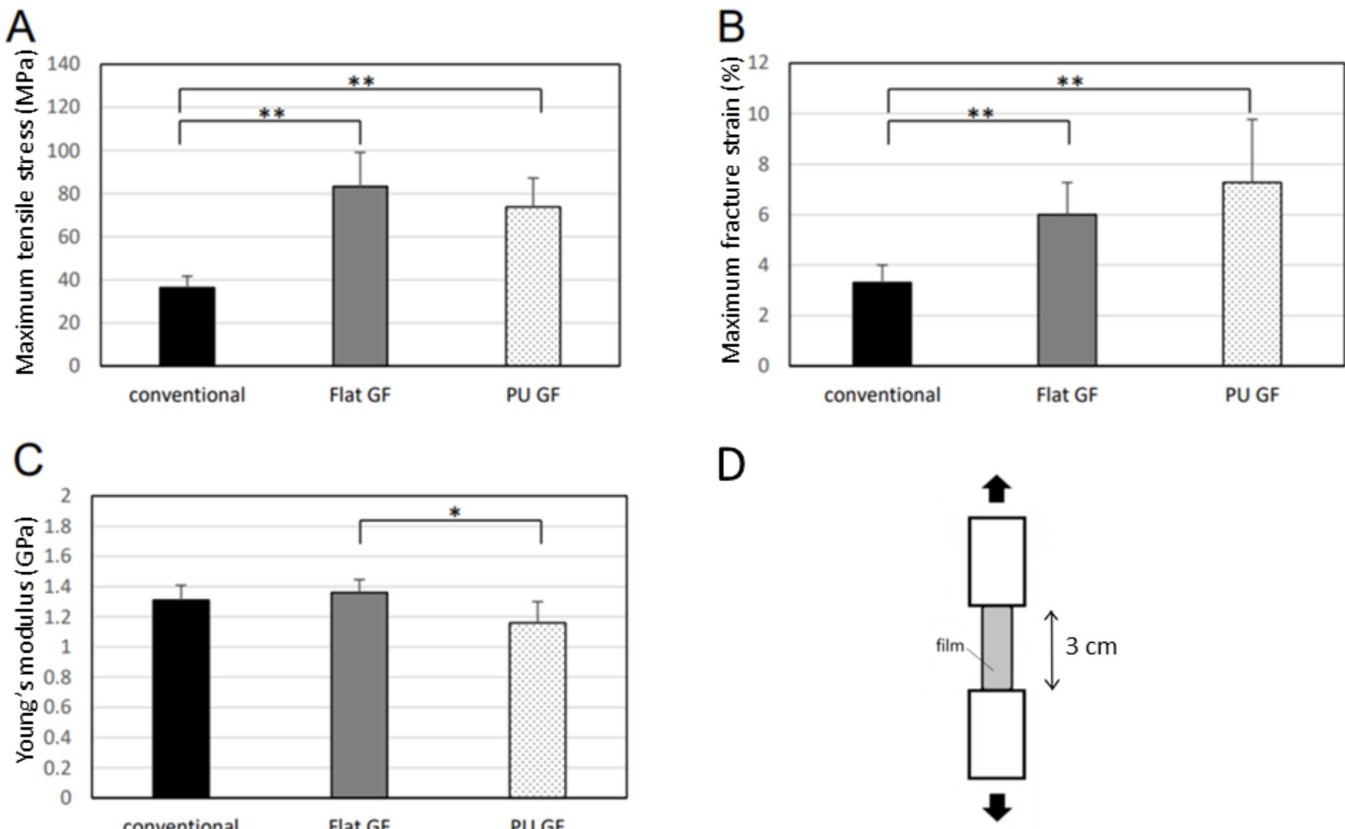

**Fig 3. The results of tensile tests.** (a) Maximum tensile stress, (b) Maximum fracture strain, (c) Young's modulus, (d) Schematic illustration of tensile test *: $p < 0.05$, **: $p < 0.01$.

lower than that of Flat GF ($p < 0.05$). Thus, PU GF has excellent physical strength and ductility.

## Adhesive test

### Pull adhesive examination

The results of the pull adhesive exam are shown in Fig 4A and 4B. All GFs were detached from the abdominal walls and were not torn during the test, whereas all conventional films tore. The maximum adhesive stresses of the conventional film, Flat GF, and PU GF were 2.16 ± 0.18, 3.67 ± 0.58, and 4.0 ± 0.76 MPa, respectively. The maximum pull adhesive stresses of Flat GF and PU GF were significantly higher than that of the conventional film ($p < 0.01$). Thus, PU GF has better tissue adhesiveness.

### Shear stress examination

The results of shear stress exam are shown in Fig 4C and 4D. All films tore during measuring. The maximum shear stresses of the conventional film, Flat GF and PU GF were 0.88 ± 0.18, 2.83 ± 0.29, and 2.2 ± 0.29 MPa, respectively. The maximum shearing stresses of Flat GF and

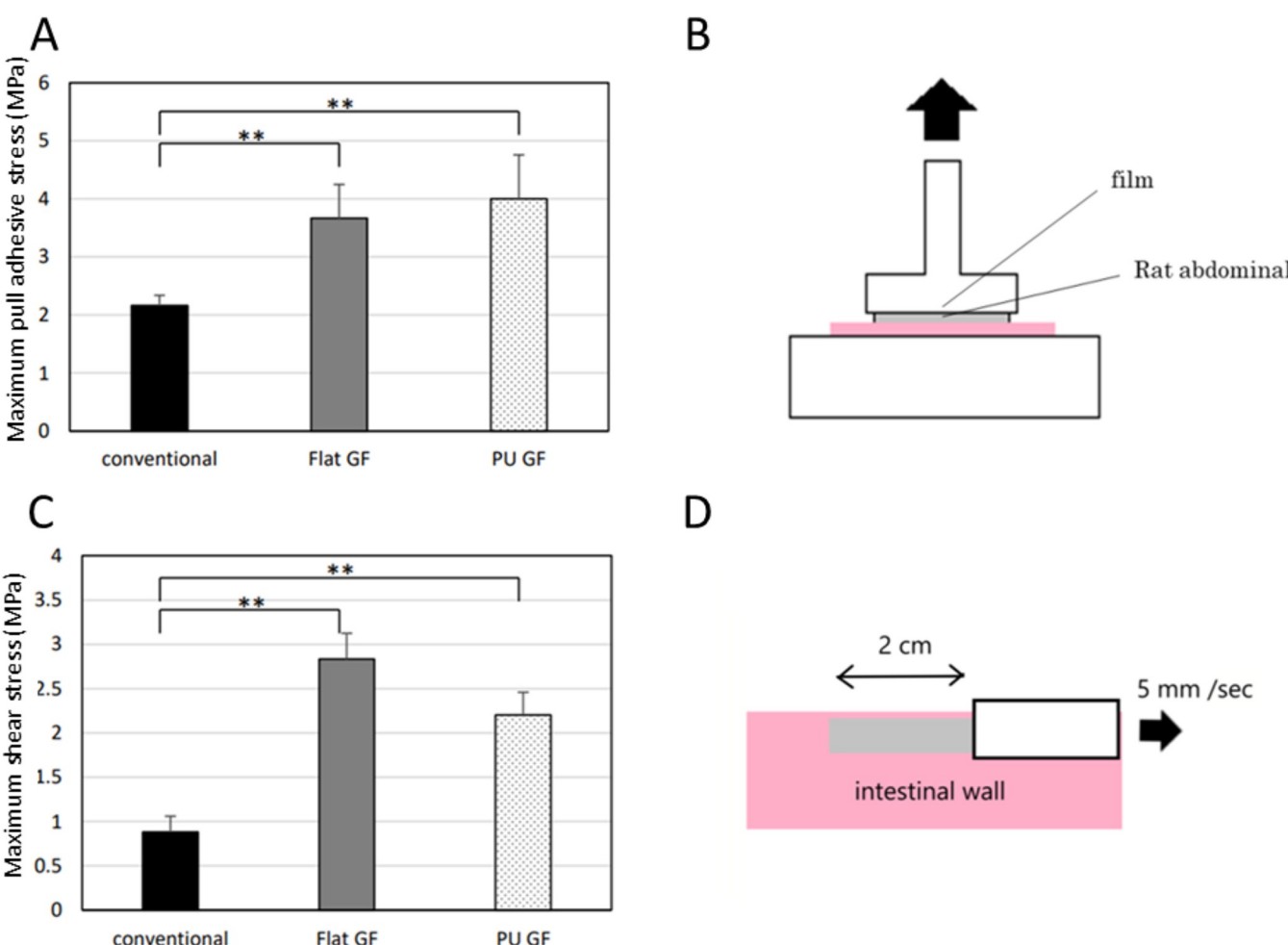

**Fig 4. The results of adhesive tests.** (a) Maximum pull adhesive stress, (b) Schematic illustration of Pull adhesive examination, (c) Maximum shear stress, (d) Schematic illustration of Shear stress examination **: $p < 0.01$.

PU GF were significantly higher than that of the conventional film ($p < 0.01$). Thus, PU GF has better tissue adhesiveness and physical strength when the film detached from abdominal tissue.

## Film-to-film attachment test

The results of the film-to-film attachment test are shown in Fig 5A and 5B. All GFs were completely separated by two edges, whereas all of the conventional films were not. The maximum film-to-film adhesive stresses of the conventional film, Flat GF and PU GF were $0.18 \pm 0.1$, $1.37 \pm 0.12$, and $0.93 \pm 0.045$ MPa, respectively. The maximum film-to-film adhesive stress of PU GF was significantly lower than that of Flat GF ($p < 0.01$). Thus, PU GF showed low attachment with the film itself.

## Film-to-silicon attachment test

The results of film-to-silicon attachment test are shown in Fig 5C and 5D. All GFs were completely detached from the silicon plate, whereas all of the conventional films were not. The

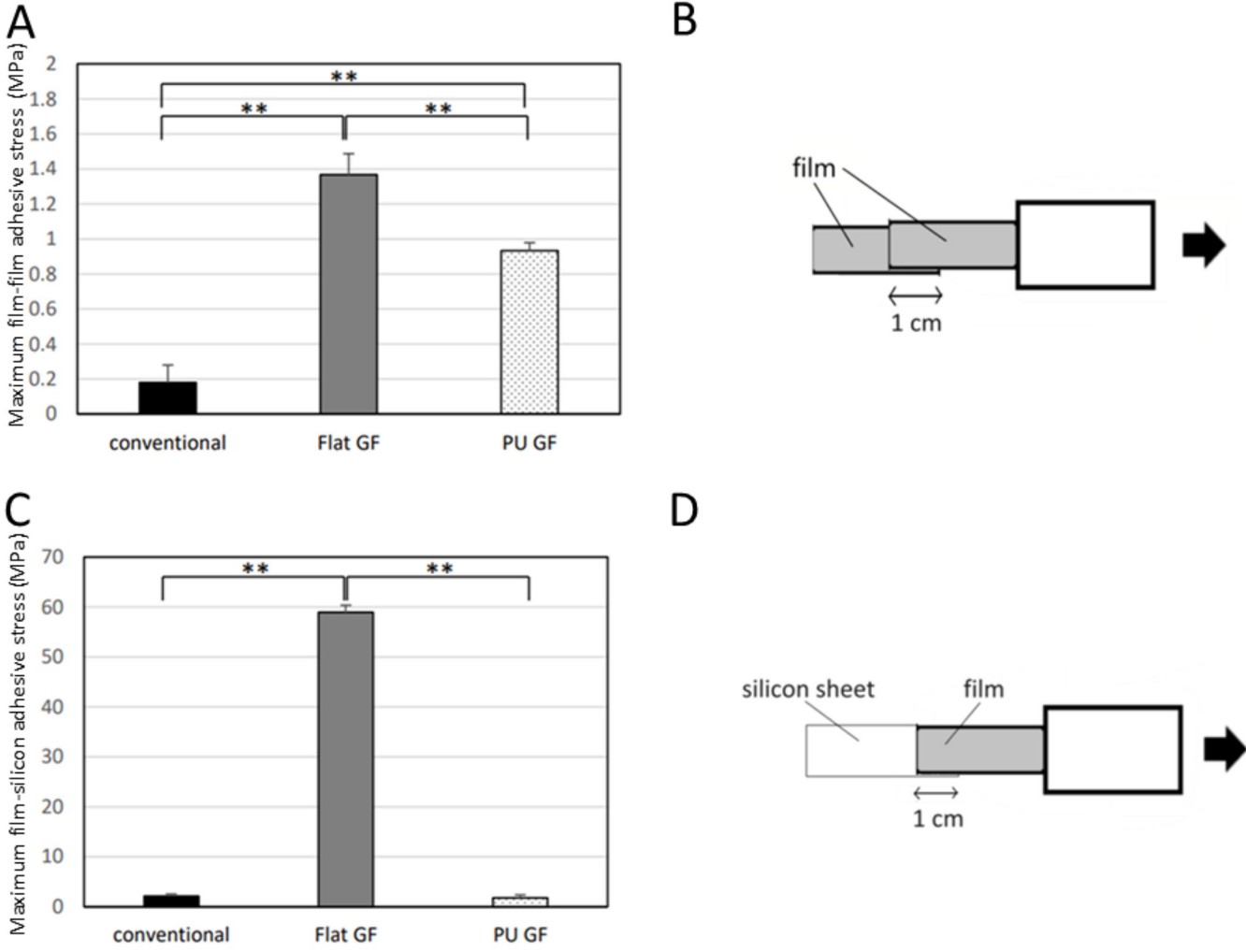

**Fig 5. The results of film attachment tests.** (a) Maximum film-to-film attachment stress, (b) Schematic illustration of Film-to-film attachment examination, (c) Maximum film-to-silicon attachment stress, (d) Schematic illustration of Film-to-silicon attachment examination **: $p < 0.01$.

maximum film-to-silicon adhesive stresses of the conventional film, Flat GF, and PU GF were 2.15 ± 0.37, 58.93 ± 1.41, and 1.77 ± 0.59 MPa, respectively. The maximum film-to-silicon adhesive stress of PU GF was significantly lower than that of Flat GF ($p < 0.01$).

## Cell proliferation test

The results of the cell proliferation tests are shown in Fig 6. The number of cells in the control at day 1, 3, 5, and 7 was 27535.45 ± 1813.04, 77161.23 ± 10505.15, 154571.26 ± 8195.47, and 201302.97 ± 16434.28. The number of cells in the conventional film at day 1, 3, 5, and 7 was 219.23 ± 10.39, 0.00 ± 0.00, 0.00 ± 0.00, and 0.00 ± 0.00. The number of cells in Flat GF at day 1, 3, 5, and 7 was 4608.43 ± 174.18, 8465.50 ± 674.79, 22899.53 ± 5948.41, and 62184.03 ± 9337.46. The number of cells in PU GF at day 1, 3, 5, and 7 was 16377.31 ± 1352.21, 23142.81 ± 9154.85, 45194.68 ± 9164.09, and 42110.54 ± 24175.39. The level of cell growth on the films was inferior to that observed in the control wells. However, the cell proliferation of PU GF and Flat GF increased with time after 1, 3, 5, and 7 days, while there was no proliferation of cells on the conventional film during all days. There were significant differences between GF and the conventional film at Days 1, 5 and 7 ($p<0.05$).

## Anti-adhesive effect test

**Single-use anti-adhesive effect examination.** The results of the anti-adhesive effect exam are shown in Fig 7A. There were no morbidities or mortalities during the operation and

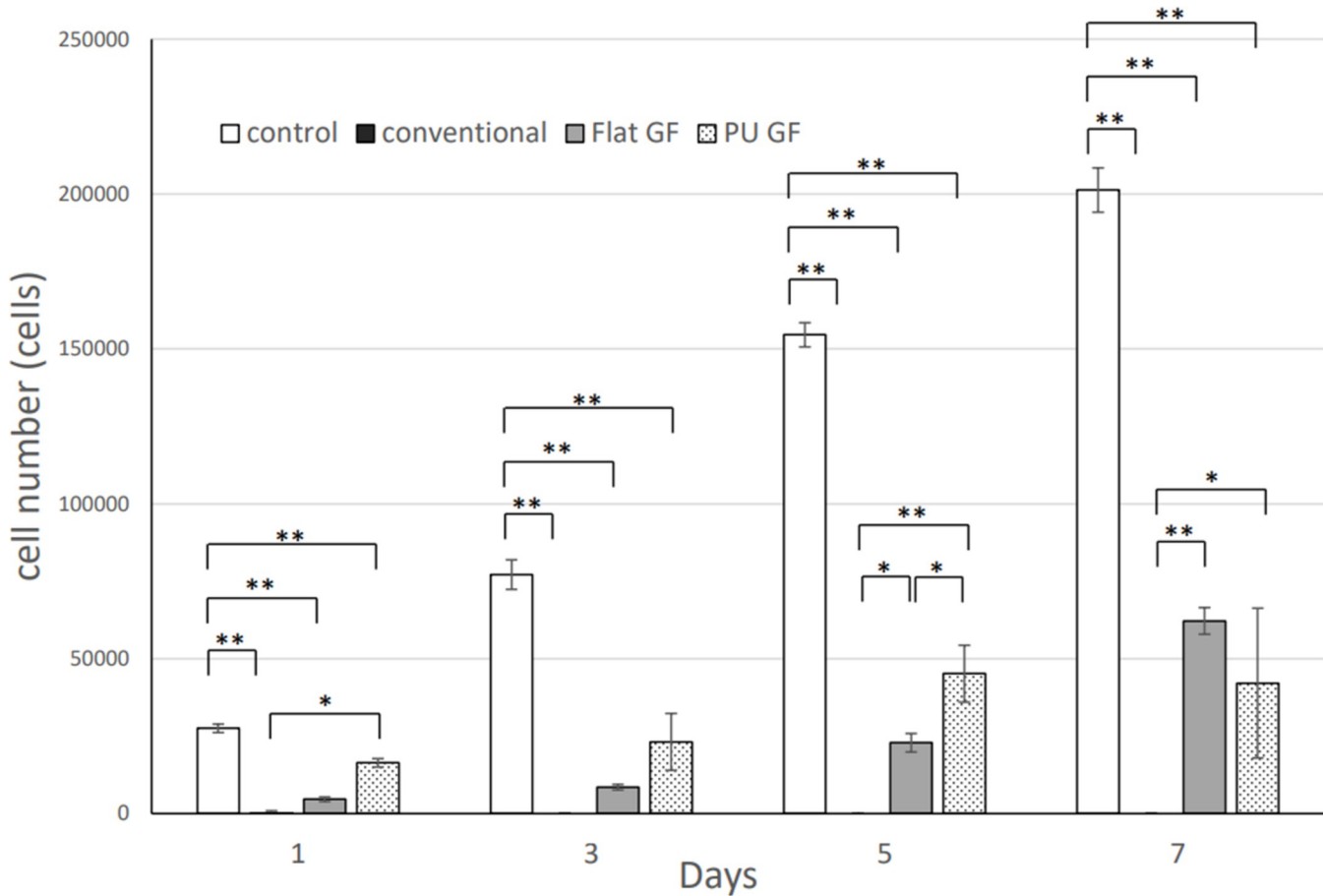

**Fig 6. The result of cell proliferation test.** *: $p < 0.05$, **$p < 0.01$.

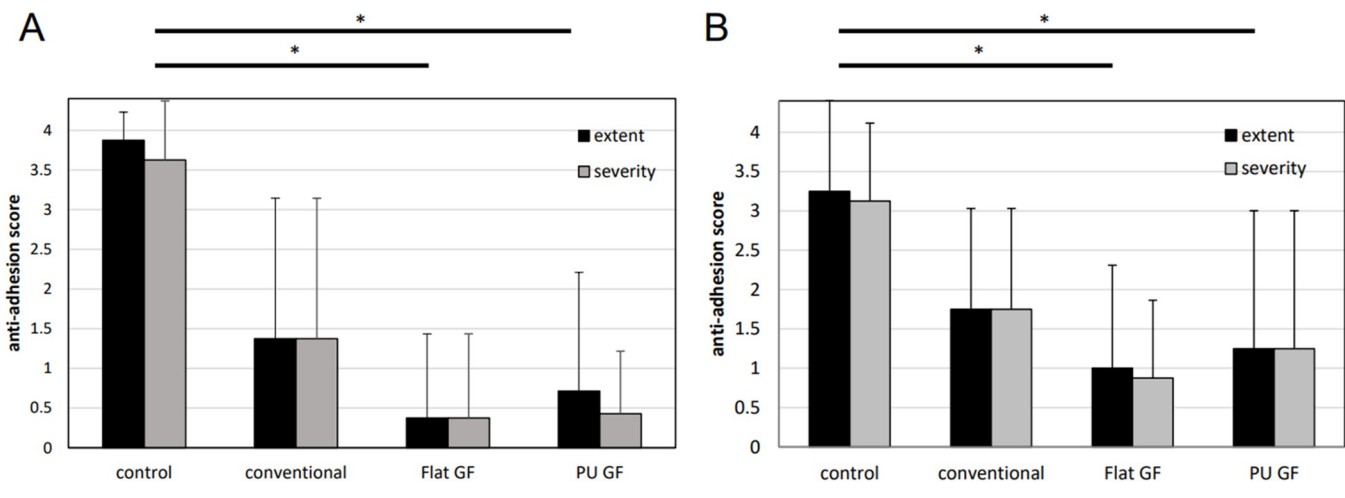

**Fig 7. The results of anti-adhesive tests.** (a) Single-use anti-adhesive scores, (b) Anti-adhesive scores of re-attachment of film *: $p < 0.05$.

observation period. There was no remaining film in the abdominal cavity in each group. The anti-adhesive scores of the control group in extent and severity were 3.88 ± 0.35 and 3.63 ± 0.74. The scores of the conventional film group in extent and severity were 1.38 ± 1.77 and 1.38 ± 1.77. The scores of the Flat GF group in extent and severity were 0.38 ± 1.06 and 0.38 ± 1.06. The scores of the PU GF group in extent and severity were 0.71 ± 1.50 and 0.43 ± 0.79. The scores of the PU GF and Flat GF groups were significantly lower than that of the control group in both extent and severity ($p < 0.05$). However, there was no significant difference between the scores of the conventional film group and the control group.

**Anti-adhesive effect examination of re-attachment of the film.** The results of anti-adhesive effect exam of re-attachment of the film and the histological images of HE staining are shown in Figs 7B and 8 There were no morbidities or mortalities during the operation and observation period. There was no remaining film in the abdominal cavity in each group. The anti-adhesive scores of the control group in extent and severity were 3.25 ± 1.16 and 3.13 ± 0.99. The scores of the conventional film group in extent and severity were 1.75 ± 1.28 and 1.75 ± 1.28. The scores of the Flat GF group in extent and severity were 1.00 ± 1.31 and 0.88 ± 0.99. The scores of the PU GF group in extent and severity were 1.25 ± 1.75 and 1.25 ± 1.75. The scores of the PU GF and Flat GF groups were significantly lower than that of the control group in both extent and severity ($p < 0.05$). However, there was no significant difference between the scores of the conventional film group and the control group. The microscopic views of control and the conventional film group showed that the abraded intestine hardly adhered to the sutured abdominal wall and numerous inflammatory cells, such as lymphocyte and macrophages. In contrast, those of PU GF and Flat GF showed thick fibrous changes of the connective tissue and inflammatory cells.

## Discussion

To improve flexibility and delivery through a laparoscopic port, anti-adhesive materials should have the physical strength, ductility, expandability, adherence to tissue and low adherence to themselves and to laparoscopic trocar. However, HA/CMC film has been problematic issue clinically because of its fragility, such as the difficulty of rolling the film and ease of adherence to laparoscopic trocar or the film itself, due to controversial clinical effectiveness for preventing adverse events of adhesion and a potential risk of anastomosis leakage at the site of

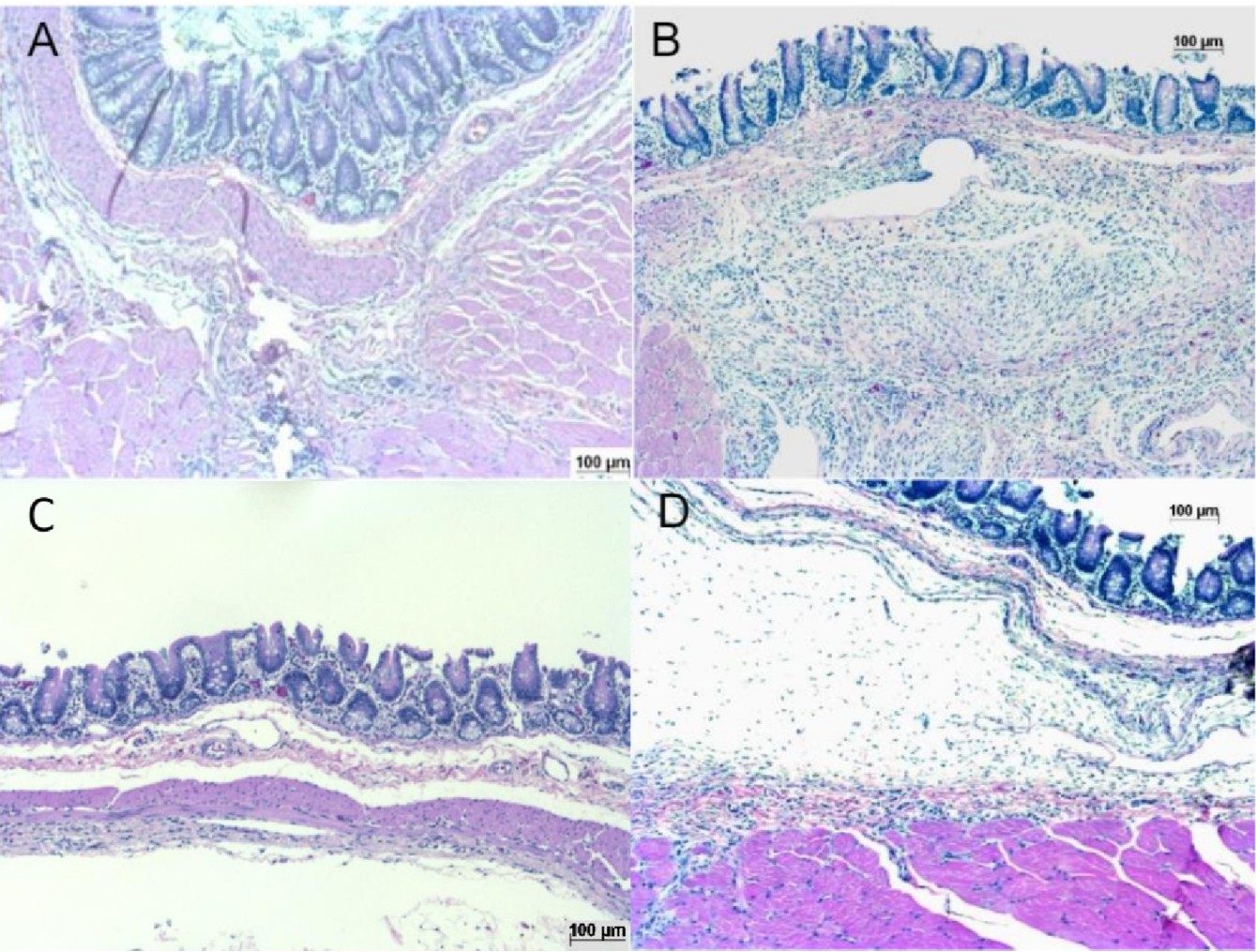

**Fig 8. Histological images of HE staining in anti-adhesion effect examination of re-attachment of the film.** (a) control, (b) conventional film, (c) Flat GF, (d) PU GF.

intestine [19, 20]. In our previous study, although it showed better physical strength, anti-adhesive effect with peritoneal regeneration than the conventional film, Flat GF had problems of laparoscopic handling such as expandability and adherence properties due to static electricity [28].

To solve these problems, we developed PU GF. PU GF decreases the contact area between the film itself and between the film and trocar. In addition, PU GF fits tissue after moistening [29]. The degradation of PU GF In vitro and in vivo can control by thermally cross-linking time, as shown in our previous study [23]. The water content of 86 ± 3% in the PU GF and Flat GF degrade for approximately one week, which is important period of anti-adhesion. In the present study, PU GF showed higher tensile strength, ductility, and adherence to tissue than the conventional film. Moreover, the film-to-film and film-to-silicon sheet adherence of PU GF were much lower than those of Flat GF. These results suggest that PU GF can provide better handling than Flat GF and the conventional films.

Before anti-adhesive effect test *in vivo*, we confirmed the anti-adhesive mechanism in cell proliferation exam on the films *in vitro*. In the results, the cell number of Flat GF, PU GF and

the conventional film were inferior than that of the control, indicating that these films may success their anti-adhesive effect by suppressive proliferation of fibroblasts on their films. In contrast, the proliferation of Flat GF and PU GF rised at each time point, while there was no proliferation in the conventional film during all culturing days. In our previous study, the extraction of gelatin film showed no cytotoxicity, and we clarified the safety, while that of the conventional film had cytotoxic effect [24]. These results indicate that the cell proliferation on Flat GF and PU GF was suppressed simply by dissolving the surface of the film, although that on the conventional film was suppressed by its cytotoxicity mainly.

Another important property for an anti-adhesive material is re-attachability, including physical strength and anti-adhesive effect. The conventional film has been used in laparoscopic surgery, however, inserting or handling using fragile film in a tight abdominal cavity is quite difficult, which increases the risk of the film making contact with incorrect sites, tearing easily, and being difficult to reuse. Therefore, we confirmed the anti-adhesive effect of PU GF after attachment only one time and on re-attachment. The anti-adhesive scores of PU GF after attachment only one time, as well as those of Flat GF, were significantly higher than those of the control. Moreover, the anti-adhesive scores of the PU GF after re-attachment, as well as those of Flat GF, were also significantly higher than those of the control. Thus, PU GF may be more useful and effective anti-adhesive material, including re-attachability, than the conventional film. Our findings suggest that PU GF improve handling properties of laparoscopic surgery as it has excellent physical strength, ductility, and adherence to tissue, and low adherence to trocar. In addition, the punctate film may be more useful with the re-attachability without tearing and to retained sufficient anti-adhesion effect.

Some limitation of this study needs to be discussed. In the film-to-silicon attachment test of physical examination, it was a pseudo-reproduction because the silicon was different from actual laparoscopic trocar. The laparoscopic handling should be examined to confirm the clinical applicability in the future study. Subsequently, though it is impossible to blind completely at the time of implantation of films, the animals were randomly allocated to a group and the evaluation was performed blind by three experts. Moreover, future studies are needed to examine the anti-adhesion effect test in large animal using a marked inflammatory model.

## Conclusion

PU GF provided better handling than Flat GF due to higher physical strength, ductility, and lower attachment to film or trocar. Moreover, PU GF had a significant anti-adhesive effect after re-attachment. These findings suggest that PU GF is useful and quite favorable as an anti-adhesive material.

## Acknowledgments

The authors would like to thank E. Nakamachi for technical support of physical tests.

## Author Contributions

**Conceptualization:** Shinichiro Morita, Hiroshi Ichikawa, Akeo Hagiwara, Masaji Tani.

**Data curation:** Shinichiro Morita, Hiroshi Minato, Joe Ueda.

**Formal analysis:** Hiroyuki Tsujimoto.

**Investigation:** Yasumitsu Oe, Joe Ueda, Akihiro Kawauchi, Susumu Kageyama.

**Methodology:** Hiroyuki Tsujimoto, Susumu Kageyama.

**Project administration:** Akeo Hagiwara.

**Resources:** Shinichiro Morita, Yasumitsu Oe.

**Software:** Yuki Ozamoto.

**Supervision:** Yuki Ozamoto, Hiroshi Ichikawa, Akihiro Kawauchi, Akeo Hagiwara, Masaji Tani.

**Visualization:** Tsunehito Horii, Hiroshi Minato.

**Writing – original draft:** Tsunehito Horii.

**Writing – review & editing:** Hiroyuki Tsujimoto.

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
