## [Decision Letter · Decision Letter 0]

21 Aug 2024

PONE-D-24-20804Physical and biological properties of a novel anti-adhesive punctate uneven gelatin film.PLOS ONE

Dear Dr. Horii,

Thank you for submitting your manuscript to PLOS ONE. After careful consideration, we feel that it has merit but does not fully meet PLOS ONE’s publication criteria as it currently stands. Therefore, we invite you to submit a revised version of the manuscript that addresses the points raised during the review process.

We look forward to receiving your revised manuscript.

Kind regards,

Rupak Dua, Ph.D.

Academic Editor

PLOS ONE

Journal Requirements:

"None of authors have any conflicts of interest to declare."

Additional Editor Comments:

Dear Dr. Horii

We have received the reports from our reviewers on your manuscript, 'Physical and biological properties of a novel anti-adhesive punctate uneven gelatin film', which you submitted to PLOS ONE

Based on the advice received, the I feel that your manuscript could be reconsidered for publication should you be prepared to incorporate major revisions.

When preparing your revised manuscript, you are asked to carefully consider the reviewer comments which are attached below, and submit a list of responses to the comments.

Reviewers' comments:

Reviewer's Responses to Questions

**Comments to the Author**

1. Is the manuscript technically sound, and do the data support the conclusions?

Reviewer #1: Yes

Reviewer #2: Yes

2. Has the statistical analysis been performed appropriately and rigorously? 

Reviewer #1: Yes

Reviewer #2: Yes

3. Have the authors made all data underlying the findings in their manuscript fully available?

Reviewer #1: Yes

Reviewer #2: Yes

4. Is the manuscript presented in an intelligible fashion and written in standard English?

Reviewer #1: Yes

Reviewer #2: Yes

5. Review Comments to the Author

Reviewer #1: 1. The abstract could be more specific in outlining the implications of the findings and how they advance current knowledge.

2. While the introduction provides a solid background, it lacks recent references that could provide a more up-to-date context for the research.

3. The methods section, although detailed, might not be sufficiently comprehensive regarding the reproducibility of the experimental setup.

4. The results section could be more integrated with the discussion to provide immediate interpretation alongside the data presented.

5. The discussion does not thoroughly address the limitations of the study.

Reviewer #2: Ozamoto et al demonstrated Gelatin films (Uneven and Flat) as anti-adhesives by comparing their characteristics with the conventionally used anti-adhesive film (i.e. Seprafilm®; Genzyme Co., Cambridge, MA, USA). The study suggests gelatin films as an interesting material with charatcerisitcs needed for anti-adhesive materials, however, the work needs clarification on the following aspects

Major:

1. Since increased cell proliferation with gelatin films might also enhance adhesions while improving regeneration, it may lead Pro- vs Anti-proliferative effects and may be clarified.

2. Biodegradability of the films among other required properties may be discussed

3. Since article has nine main Figs and seem to need reorganization E.g. each test schematic (in Fig.2) may be accompany the data and may be kept as a separate Fig

Minor:

1. Composition of gelatin other than collagen-I may be mentioned for healing perspective

2. Overall, Figs quality may be improved

3. Fig.1 Surface roughness measured for dried. Is there any range when hydrated?

4. Fig labels may be updated (e.g. "Maximum" tensile stress In Fig.4)

5. Error bars indicate SD or SEM (Number of samples may be indicated)?

6. PLOS authors have the option to publish the peer review history of their article (what does this mean?). If published, this will include your full peer review and any attached files.

Reviewer #1: No

Reviewer #2: No

---

## [Author Response · Author response to Decision Letter 0]

24 Sep 2024

1.manuscript meets PLOS ONE's style

(Answer)

Following to your manuscript guideline, I changed and corrected the contents of our manuscript.

2.in your Methods section, please provide additional information regarding the experiments involving animals and ensure you have included details on (1) methods of sacrifice, (2) methods of anesthesia and/or analgesia, and (3) efforts to alleviate suffering.

(Answer)

Corresponding to your comment, I added the methods of animal anesthesia, efforts to alleviate suffering and sacrifice in the methods section as Follows, “Large intestinal walls (length of 15 cm) were resected from three beagle dogs under intravenous anesthesia of pentobarbital (40 mg/kg of body weight) and inhalation anesthesia of sevoflurane (1-2 %).”, “Under intraperitoneal administration of pentobarbital (40 mg/kg of body weight) and inhalation anesthesia of sevoflurane (1-2 %),”, “all animals were sacrificed by intraperitoneal administration of pentobarbital (100 mg/kg of body weight)”.

3.Please complete your Competing Interests on the online submission form to state any Competing Interests. If you have no competing interests, please state "The authors have declared that no competing interests exist."

(Answer)

We have no competing interests. Thus, I corrected the statement about it in acknowledgements.

4.Please confirm at this time whether or not your submission contains all raw data required to replicate the results of your study.

(Answer)

Following to your comment, I checked all data in our manuscript and added the raw data in the cell proliferation test and anti-adhesive tests.

5.Please ensure that you have an ORCID iD and that it is validated in Editorial Manager.

(Answer)

I confirmed my ORCID iD linked to profile in Plos one online submission page.

6.Please include captions for your Supporting Information files at the end of your manuscript, and update any in-text citations to match accordingly.

(Answer)

Corresponding to your comment, I included the captions at the end of our manuscript.

Additional Editor comments:

Reviewer1：

1.The abstract could be more specific in outlining the implications of the findings and how they advance current knowledge.

(Answer) 

Following to your comment, I changed and added the comment in the abstract as follows, “Our findings suggest that PU GF improve handling properties of laparoscopic surgery as it has excellent physical strength, ductility, and adherence to tissue, and low adherence to trocar. In addition, the punctate film may be more useful with the re-attachability without tearing and to retained sufficient anti-adhesion effect.”

2. While the introduction provides a solid background, it lacks recent references that could provide a more up-to-date context for the research.

(Answer)

Corresponding to your comment, I added recent references about anti-adhesion and HA/CMC anti-adhesive materials in the Introduction section.

3. The methods section, although detailed, might not be sufficiently comprehensive regarding the reproducibility of the experimental setup.

(Answer)

As for the reproducibility of the experimental setup in the methods section, the experimental methods of tensile test and pull adhesive test are based on ISO527-3 and our previous report (Horii T et al., JBMR-B 2018). Also, the methods of shear stress examination, film-to-film attachment examination and film-to-silicon attachment examination are based on imitated EN1939 and our previous report (Horii T et al., JBMR-B 2018). Subsequently, the methods of cell proliferation test and anti-adhesive tests are based on our previous reports (Horii T et al., JBMR-B 2018, Tsujimoto H et al., JBMR-B 2015, Miyamoto H et al., JBMR-B 2018).

4. The results section could be more integrated with the discussion to provide immediate interpretation alongside the data presented.

(Answer)

Corresponding to your comment, I added the comments in the result as follows, “Thus, PU GF has excellent physical strength and ductility.”, “Thus, PU GF has better tissue adhesiveness.”, “Thus, PU GF has better tissue adhesiveness and physical strength when the film detached from abdominal tissue.”, “Thus, PU GF showed low attachment with the film itself.”, “Thus, PU GF also showed low attachment with silicon tube.”

5. The discussion does not thoroughly address the limitations of the study.

(Answer)

Following to your comment, I mentioned the limitations of the study in discussion as follows, “Some limitation of this study needs to be discussed. In the film-to-silicon attachment test of physical examination, it was a pseudo-reproduction because the silicon was different from actual laparoscopic trocar. The laparoscopic handling should be examined to confirm the clinical applicability in the future study. Subsequently, though it is impossible to blind completely at the time of implantation of films, the animals were randomly allocated to a group and the evaluation was performed blind by three experts. Moreover, future studies are needed to examine the anti-adhesion effect test in large animal using a marked inflammatory model.”

Reviewer2：

Major:

1. Since increased cell proliferation with gelatin films might also enhance adhesions while improving regeneration, it may lead Pro- vs Anti-proliferative effects and may be clarified.

(Answer)

In terms of the relationship between the cell proliferation on the gelatin film and anti-adhesive effect, we previously reported that fibroblast proliferation on gelatin film was suppressed simply by dissolving the surface of the film, because the gelatin film extract had no cytotoxicity. In contrast, fibroblast proliferation on the conventional film was suppressed by not only dissolving the surface of the film but also due to its cytotoxicity. Thus, the mild cell proliferation on the gelatin film and high safety of the extract of the film effect excellent anti-adhesive effect with peritoneal regeneration.

2. Biodegradability of the films among other required properties may be discussed

(Answer)

Biodegradability of the PUGF was predicted by the cross-linking time, regarding as our previous report. (H Tsujimoto et al., J Biomed Res Part B, 2014) It was revealed that the Flat GF (3.5 hrs cross-linking time, 140 oC) degrades for 7 days in vivo degradation study. I’ve already discussed about the degradation of GF in discussion as follows, “The degradation of PU GF In vitro and in vivo can control by thermally cross-linking time, as shown in our previous study [23]. The water content of 86 ± 3% in the PU GF and Flat GF degrade for approximately one week, which is important period of anti-adhesion.”

3. Since article has nine main Figs and seem to need reorganization E.g. each test schematic (in Fig.2) may be accompany the data and may be kept as a separate Fig

(Answer)

Corresponding to the comment, Figures are reorganized, especially Fig. 2 was separated and accompany the data with schematic illustrations.

Minor:

1. Composition of gelatin other than collagen-I may be mentioned for healing perspective

(Answer)

Actually, as composition of gelatin, only alkali-treated gelatin without telopeptide was used and any other cross-linking agent was not used. Also, the gelatin has low endotoxin.

2. Overall, Figs quality may be improved

(Answer)

Following to the comment, the quality of all Figures was improved.

3. Fig.1 Surface roughness measured for dried. Is there any range when hydrated?

(Answer)

As for the surface changes of PUGF on wet condition, there was not any change when it moistened with distilled water using a spray, positioned 5 cm from the film and pushed 5 times. Thus, I added the comment about it as follows, “We confirmed that there was almost no change of the surface structure of the PU GF in wet condition just after moisture.”

4. Fig labels may be updated (e.g. "Maximum" tensile stress In Fig.4)

(Answer)

Following to the comment, I modified the Figure labels.

5. Error bars indicate SD or SEM (Number of samples may be indicated)?

(Answer)

Error bars indicate SD. Thus, I added the comments in materials and methods as follows,” Continuous variables are presented as mean ± standard deviation.”

---

## [Decision Letter · Decision Letter 1]

6 Nov 2024

Physical and biological properties of a novel anti-adhesive punctate uneven gelatin film.

PONE-D-24-20804R1

Dear Dr. Horri

We’re pleased to inform you that your manuscript has been judged scientifically suitable for publication and will be formally accepted for publication once it meets all outstanding technical requirements.

Kind regards,

Rupak Dua

Academic Editor

PLOS ONE

Additional Editor Comments (optional):

Reviewers' comments:

Reviewer's Responses to Questions

**Comments to the Author**

1. If the authors have adequately addressed your comments raised in a previous round of review and you feel that this manuscript is now acceptable for publication, you may indicate that here to bypass the “Comments to the Author” section, enter your conflict of interest statement in the “Confidential to Editor” section, and submit your "Accept" recommendation.

Reviewer #2: All comments have been addressed

2. Is the manuscript technically sound, and do the data support the conclusions?

Reviewer #2: Yes

3. Has the statistical analysis been performed appropriately and rigorously? 

Reviewer #2: Yes

4. Have the authors made all data underlying the findings in their manuscript fully available?

Reviewer #2: Yes

5. Is the manuscript presented in an intelligible fashion and written in standard English?

Reviewer #2: Yes

6. Review Comments to the Author

Reviewer #2: Ozamoto et al seems to have addressed most of my concerns in the revised version of the manuscript.

7. PLOS authors have the option to publish the peer review history of their article (what does this mean?). If published, this will include your full peer review and any attached files.

Reviewer #2: No

---

## [Editor Report · Acceptance letter]

13 Dec 2024

PONE-D-24-20804R1 

PLOS ONE

Dear Dr. Horii, 

I'm pleased to inform you that your manuscript has been deemed suitable for publication in PLOS ONE. Congratulations! Your manuscript is now being handed over to our production team.

Kind regards, 

on behalf of

Dr. Rupak Dua 

Academic Editor

PLOS ONE